# One-Pot Synthesis of Dual Color-Emitting CDs: Numerical and Experimental Optimization towards White LEDs

**DOI:** 10.3390/nano13030374

**Published:** 2023-01-17

**Authors:** Gianluca Minervini, Antonino Madonia, Annamaria Panniello, Elisabetta Fanizza, Maria Lucia Curri, Marinella Striccoli

**Affiliations:** 1Department of Electrical and Information Engineering, Polytechnic of Bari, Via E. Orabona 4, 70126 Bari, Italy; 2CNR-IPCF Bari Division, c/o Chemistry Department, University of Bari “Aldo Moro”, Via Orabona 4, 70126 Bari, Italy; 3Department of Chemistry, University of Bari “Aldo Moro”, Via Orabona 4, 70126 Bari, Italy

**Keywords:** carbon dots, solvothermal synthesis, green precursors, polymer passivation, multiple emission bands, colorimetric numerical simulation, polymer nanocomposite, white LEDs

## Abstract

Carbon Dots (CDs) are fluorescent carbon-based nanoparticles that have attracted increasing attention in recent years as environment-friendly and cost-effective fluorophores. An application that can benefit from CDs in a relatively short-term perspective is the fabrication of color-converting materials in phosphor-converted white LEDs (WLEDs). In this work we present a one-pot solvothermal synthesis of polymer-passivated CDs that show a dual emission band (in the green and in the red regions) upon blue light excitation. A purposely designed numerical approach enables evaluating how the spectroscopic properties of such CDs can be profitable for application in WLEDs emulating daylight characteristics. Subsequently, we fabricate nanocomposite coatings based on the dual color-emitting CDs via solution-based strategies, and we compare their color-converting properties with those of the simulated ones to finally accomplish white light emission. The combined numerical and experimental approach can find a general use to reduce the number of experimental trial-and-error steps required for optimization of CD optical properties for lighting application.

## 1. Introduction

Carbon Dots (CDs) are zero-dimensional nanometric particles with carbon-based chemical composition and intense fluorescence in the visible spectral region [1,2]. Thanks to their environment-friendly, heavy-metal and rare-earth free structure, CDs have been the object of extensive interest for implementation in optoelectronic devices, for substitution of more harmful and/or energy- and cost-demanding materials [3,4,5]. In particular, many recent works have focused on the use of CDs in phosphor-converted, white-emitting LEDs (WLEDs) [5,6]. In these types of devices, UV or blue light emitted from a primary solid-state semiconductor LED chip is partially absorbed and re-emitted as photoluminescence (PL) by a phosphor material so that a white light spectrum is produced as result [6,7,8,9,10]. For a given primary LED chip, the characteristics of the generated white light are dependent on the spectroscopic properties of the phosphor materials. Therefore, the PL properties of such phosphors must be opportunely controlled in order to obtain white light with suitable characteristics. In particular, a sufficiently warm light with Correlated Color Temperature (CCT) lower than 7000–7500 K and a Color Rendering Index (CRI) of at least 80 are considered as strict requirements to making the final devices suitable for application in lighting of indoor and outdoor environments [11].

For the fabrication of CD-based color-converting components, it is thus necessary to have carbonaceous nanoparticles emitting with high efficiency in the regions of blue, green and red [12,13,14,15,16,17]. Furthermore, such CDs need to be uniformly incorporated into solid-state matrices without relevant degradation of their spectroscopic characteristics [18,19,20]. In the past few years, synthesis of CDs through solvothermal treatment of various types of molecular carbonaceous precursors has emerged as a very effective approach to obtaining multicolored emissive CDs [13,21,22,23,24,25,26,27,28,29,30,31,32,33]. Instead, most of the reported methods for fabricating color-converting nanocomposites rely on blending CDs obtained through separate syntheses, having emission in the blue, green and red upon excitation by a primary source [13,21,22,23,24,25,26,27,28,29]. Although these strategies allow fine adjustment of the final CCT and CRI via easy regulation of the fluorophore’s relative concentrations, they necessarily involve multiple steps for the synthesis and purification of individual CDs. Notably, this also implies that the overall process of nanocomposite fabrication is affected by high energy consumption and large production of solvent waste. Therefore, some alternative approaches have been proposed, aiming at achieving white light relying on one-pot synthetic procedures of CDs [30,31,32,34,35,36,37,38,39,40,41,42]. In such studies, the developed strategies to fabricate white-emitting nanocomposites can be classified in three categories: (i) single-step synthesis of CDs with multiple-emitting states, having several fluorescence bands in the visible range; (ii) exploitation of concentration-dependent emission properties of CDs; (iii) control over aggregation in CDs’ nanosystems in order to obtain multiple visible emission bands [30,31,32,34,35,36,37,38,39,40,41,42]. Such one-pot methods are capable of drastically reducing the number of steps required in the overall nanocomposite fabrication process; however, controlling the properties of the final white light is harder with such approaches. For example, for some one-pot synthesized CDs, the lack of a sufficiently intense red component in the total spectrum results in cool white light with low CRI [43,44]. Moreover, problems related to an excessively low photoluminescence quantum yield (PLQY), or to aggregation-induced quenching of the fluorescence of CDs in the nanocomposites, can hinder the attainment of high color-converting performances [30,35,40,42]. 

Further, the design and development of CDs for one-pot fabrication of color-converting nanocomposites can also benefit from the availability of algorithms for predicting and optimizing the properties of the final white light color (CIE chromaticity coordinates, CCT and CRI). More specifically, the ability of correlating the optical properties of CDs resulting from the synthesis to the color-converting performances in the final devices can speed up the color converter design, by eliminating the necessity of testing every synthesized nanoparticle batch into a device to experimentally obtain the white light colorimetric properties. Presently, the use of such types of algorithms is prevalently limited to the case of separately synthesized fluorophores, for the optimization of blue, green and red components in the final spectrum. Moreover, most of these calculation methods have been conceived for the optimization of narrow emitters, namely rare-earth phosphors [45,46,47,48,49,50,51,52]. More recently, similar algorithms have been applied to color-converting components based on semiconductor quantum dots [49,53,54]. However, a computational approach explicitly taking into account specific spectroscopic properties of CDs has not yet been reported, to the best of our knowledge.

In this work, we report a one-pot solvothermal synthesis of polymer-passivated CDs that are resistant to aggregation-induced quenching of fluorescence. Indeed, surface passivation of CDs with polymer chains is an established method to enhance their PL and prevent aggregation-induced quenching [55,56,57]. Upon blue light excitation, the as-synthesized CDs show an emission spectrum with a green and a red band of comparable intensity. The capability of such CDs to act as color converters with suitable performances for daylight WLEDs is evaluated by means of a purposely designed numerical calculation method. Then nanocomposite coatings are prepared in a wide range of thicknesses and CDs contents, by incorporating the CDs in a polyvinyl alcohol (PVA) matrix, through common solution-based deposition techniques, such as drop-casting and spin-coating. Then we identify the most suitable samples for implementation of white daylight LEDs (i.e., the samples with CCT closer to 6500 K and highest possible CRI), resulting in color-converting nanocomposites with CIE chromaticity coordinates, CCT and CRI of (0.30, 0.34), 7100 K and 77 (spin-coating deposition) and of (0.30, 0.36), 6902 K and 82 (drop-casting deposition). Our work integrates numerical and experimental investigation of the colorimetric properties of the CDs’ nanocomposites prepared with step-saving fabrication procedures, demonstrating the effectiveness of the method and its general applicability for assessing the colorimetric properties of color-converting CDs-based coatings also for other lighting applications. 

## 2. Materials and Methods

### 2.1. Chemicals

Citric acid (anhydrous, CA, ≥99.5%), urea (99.0–100.5%), poly(ethyleneimine) (PEI, average M_w_ ~2000, solution 50 wt.% in H_2_O), N,N-dimethylformamide (DMF, ≥99.8%), sodium hydroxide (NaOH, ≥97%), hydrochloric acid (HCl, 37%), polyvinylpyrrolidone (PVP, K30) and Poly(vinyl alcohol) (PVA, M_w_ 9000–10,000, 80% hydrolyzed) were purchased from Sigma Aldrich, Milano, Italy, and used as received, without any further purification. All aqueous solutions were prepared using MilliQ water. 

### 2.2. Synthesis of Fluorescent CDs

Fluorescent carbon nanoparticles were prepared starting from a classical synthesis already reported in the literature [58,59] based on the thermal decomposition in autoclave of citric acid and urea dissolved in DMF. In our work, PEI is additionally introduced among the precursors as an in-situ surface passivation agent to enhance the emission properties. 

In short, 1.8 g of citric acid, 3.6 g of urea (molar ratio of 1:7) and 0.014 g of PEI (0.02 mM concentration) were dissolved into 18 mL of DMF. The resulting optically clear DMF solution was then transferred into a 50 mL autoclave lined with borosilicate glass and heated at 160 °C for 6 h. At the end of the solvothermal treatment, the resulting raw product appeared to be dark-brown and completely opaque, indicating a successful carbonization process.

To remove possible unreacted precursors and any fluorescent byproduct, the sample was purified by numerous cycles of washing/precipitation steps under centrifugation. The performed procedure was based on the surface-charge dependency of the obtained nanoparticles on the acidity of the surrounding environment, a property common to many CDs and often reported in the literature [14,60]. Our CDs precipitated after HCl addition, while remaining highly stable in basic environments. As such, a 5% HCl solution was initially added to the crude product. The acidified mixture was then centrifuged to collect the precipitate that is afterwards progressively purified through several cycles of washing with 0.5% HCl water solution and successive centrifugation. Also, other fluorescent byproducts formed in the sample mixture; these blue- and green-emitting species display similar stability in acid as the red-emitting CDs of interest. Thus, at each washing step only a small fraction of such byproducts was removed along with the supernatant. The purification was followed by recording the absorption spectrum of each supernatant (Appendix A), which showed the progressive disappearance of the unwanted components. Finally, the precipitate was dispersed in a 0.025 mol/L NaOH aqueous solution. The purified product was then freeze-dried and stored for further uses.

### 2.3. Transmission Electron Microscopy Investigation of Synthesized CDs

Transmission Electron Microscopy (TEM) analysis was carried out using a JEOL JEM-1011 microscope (JEOL, Akishima, Tokyo, Japan), equipped with a W filament operating at 100 kV. Micrographs were acquired using an Olympus Quemesa high resolution CCD camera. The samples were prepared by dipping carbon-coated copper grids in diluted aqueous solutions of CDs and then leaving the grids to dry under air. The statistical analysis on CD size was performed by using a free image analysis software (ImageJ, v.1.52a).

### 2.4. Spectroscopic Investigation

The chemical composition of synthesized CDs was investigated with Fourier-Transform Infrared (FT-IR) spectroscopy, using a PerkinElmer Spectrum One Fourier Transform Infrared spectrophotometer (PerkinElmer, Inc., Waltham, MA, USA) with the attenuated total reflection technique, using a 4-mm-diameter diamond microprism as an internal reflection element. 

UV–Vis absorption spectra were recorded with a Cary 5000 spectrophotometer (Agilent Technologies, Inc., Santa Clara, CA, USA). Steady-state photoluminescence excitation (PLE) and emission (PL) spectra of CDs solution and nanocomposites were acquired using a Fluorolog 3 spectrofluorometer (HORIBA Jobin-Yvon GmbH, Bensheim, Germany), equipped with double-grating excitation and emission monochromators and a 450W Xe lamp as light source. Absolute PLQY was measured using a “Quanta-phi” integration sphere coated with Spectralon^®^ (HORIBA Jobin Yvon GmbH, Bensheim, Germany) (reflectance ≥95% in the range 250–2500 nm). Time-Resolved PL (TRPL) measurements were carried out by the Time Correlated Single Photon Counting (TCSPC) technique, with a FluoroHub (HORIBA Jobin-Yvon). CDs solutions were excited using 80 ps laser diode sources at 375 nm (NanoLED 375L) and at 485 nm (NanoLED 485L), with a final resolution time of 300 ps. 

### 2.5. Preparation of CDs Powder Fluorophores

Fluorescent powders of CDs were fabricated starting from already reported procedures [61,62]. The purified CDs pellet obtained at the end of the purification process (Section 2.2) was hygroscopic, thus making it difficult to handle a controllable amount of embedded CDs powders in nanocomposite preparation. Therefore, a defined amount of CDs was embedded in PVP to obtain a chemically stable fluorescent powder, easily processable at ambient conditions. In details, the purified CDs (2 mg) were dissolved in MilliQ water (1 mL) in presence of PVP (0.4 g) under vigorous magnetic stirring. The resulting viscous solution was drop-cast onto the surface of a glass plate where water was left to evaporate firstly at room temperature (RT) for ~1 h and then in an oven at 60 °C for 2–3 h, forming a compact deposited layer. Then the solid film was scratched off the glass surface and finely ground into a mortar to obtain an emitting CD powder. In the next section, we refer to “CD powder” as the ground solid PVP-CDs prepared as reported here. 

### 2.6. Preparation of CDs Nanocomposite Films

Nanocomposite films were prepared by embedding the previously prepared CD powder in a PVA polymeric matrix, as depicted in Figure 1. In details, CD solid-state fluorophores were added in variable amounts (2–120 mg/mL of CD powder) to aqueous solutions of PVA (0.2–0.4 g/mL) and kept under magnetic stirring until obtaining optically clear dispersions. Such transparent dispersions, hereinafter referred to as “inks”, were used to deposit fluorescent nanocomposite films onto glass or quartz substrates.

Two different techniques were used to obtain nanocomposite films, namely drop-casting and spin-coating, depending on the desired film thickness. In the first case, the nanocomposite solutions were deposited onto a quartz substrate (10 mm × 10 mm × 1 mm) by using a drop volume of either 50 or 100 μL and leaving the solvent to evaporate. Specifically, in case of inks with 0.4 g/mL PVA, the solvent was allowed to evaporate slowly at RT (~20 °C). For 0.2 g/mL PVA inks, the films were kept on a heating plate at 40 °C to accelerate the solvent evaporation. 

Nanocomposite solutions were deposited by spin-coating onto same size quartz substrates by using a two-step spinning recipe: 30 s at 800–1000 rpm (low speed) and 60 s at 6000 rpm (high speed) to ensure an effective edge bead removal.

### 2.7. Scanning Electron Microscopy Investigation of Prepared Nanocomposites

In-plane and cross-section Scanning Electron Microscopy (SEM) micrographs of the prepared nanocomposites were acquired using a Zeiss Sigma microscope (Carl Zeiss Co., Oberkochen, Germany) operating in the range 0.5–20 kV and equipped with an in-lens secondary electron detector. FE-SEM micrographs were recorded by mounting the nanocomposite films onto stainless-steel sample holders by using double-sided carbon tape. 

An estimation of film thickness was obtained from cross-section SEM micrographs (d_SEM_) on the vertically aligned samples by measuring the film thickness in several central positions and computing the average value. The error of such thickness was calculated as the root mean square deviation from the average.

### 2.8. Analysis of PL Spectra for Measurement of Colorimetric Properties

Chromaticity Coordinates (CIE 1931 standard 2-degree observer), Correlated Color Temperature (CCT) and general Color Rendering Index (CRI) were obtained using MATLAB software (R2021b, v. 9.11.0), with the Add-on pspectro [63]. Relative spectral density of irradiances (normalized 1 nm-spaced PL spectra in absolute radiometric units, W·cm^−2^) were used as input for colorimetric property calculation. Relative spectral density of irradiances resulted either from numerical spectral simulation or from experimental colorimetric measurements, as described in the following sections.

### 2.9. Numerical Estimation of Best Achievable White Colorimetric Properties

Simulated relative spectral densities of irradiances were obtained from a linear combination of external LED chip emission spectrum (blue component) and PL of CDs (green/red component), as described in Equation (1). Best white chromaticity coordinates were calculated as the simulated (x, y) values that minimize the function:(1)$${\left(x_{i}{\;-\;x}_{D65}\right)}^{2}+{\left(y_{i}{\;-\;y}_{D65}\right)}^{2}$$
where x_i_ and y_i_ are the simulated sample chromaticity coordinates and (x_D65_, y_D65_) = (0.31272, 0.32903) are the chromaticity coordinates of D65 illuminant, selected as reference standard, following CIE recommendations on uniformity of practice in the evaluation of whiteness of surface colors [11]. The simulated spectrum correspondent to the best achievable (x, y) is then used as input to calculate the best achievable CCT and CRI. 

### 2.10. Experimental Measurement of Nanocomposite Colorimetric Properties

Colorimetric properties of nanocomposites were measured by mounting opportunely the nanocomposite films on quartz substrates in the sample chamber of a Fluorolog 3 spectrofluorometer (HORIBA Jobin-Yvon GmbH, Bensheim, Germany). A Kingbright^®^ Blue (InGaN) LED (peak wavelength: 465 nm) was used as primary blue light source, and specific precautions were taken to prevent presence of stray incident light or reflected light. Resulting spectra were converted in relative spectral density of irradiances (by normalizing and expressing the spectra in absolute radiometric units, W cm^−2^) and used as input to calculate chromaticity coordinates, CCT and CRI, by applying the procedure reported in Section 2.8.

## 3. Results and Discussion

### 3.1. Characterization of CDs Fluorophores

The morphological characterization of CDs was performed by TEM (Figure 1a); the obtained micrographs show nanoparticles with average diameter of 2.5 nm (σ = 0.8 nm). In some TEM grid regions, aggregates of CDs (sized ~10 nm) can be found, which may be ascribed to the presence of the PEI passivating agent inducing aggregation of individual CDs due to cross-linking of the polymer chains and/or to water evaporation during TEM grid preparation (Appendix A). Though many reports state that aggregation usually induces a quenching of the CDs’ fluorescence [14,64], herein such a behavior was not observed for the passivated synthesized nanoparticles (as shown in the following) thus resulting amenable for the preparation of solid-state fluorescent nanocomposites based on such CDs.

DLS analysis (Figure 1b) indicates a CDs’ size of 66.6 nm (σ = 27.3 nm). The discrepancy of this size value with that resulting from TEM investigation can be ascribed to the hydrodynamic radius measured by DLS that takes into account both the presence of organic groups on the CD surface, possibly formed during the synthetic process, and the presence of the passivating layer of PEI enveloping the nanoparticles. Such organic shell on the surface could hardly be observed by TEM due to its significantly low contrast. Overall, the nanoparticles appear composed by a carbonaceous core, clearly visible in the TEM micrographs, and a softer surface shell is evidenced by the DLS measurements.

Composition of the surface shell of the synthesized CDs was thus studied via FT-IR spectroscopy (Figure 1c). The complex and feature-rich spectrum observed for CDs is indicative of a shell composed by many different organic moieties. It is thus possible that PEI is able to envelop the CDs by interacting with pre-existing functional surface groups. The very broad band observed around 3340 cm^−1^ can be ascribed to O–H stretching vibrations; at 3200 cm^−1^ it is possible to distinguish another broad band, which is associated to N–H stretching, and a triplet of peaks associated to C–H stretching (2960 cm^−1^, 2925 cm^−1^, and 2850 cm^−1^). The presence of amide groups on the CDs surface is confirmed by the presence of the C=O stretching peak (amide I band) found at 1650 cm^−1^, of the N–H_2_ deformation peak (amide II band) at 1550 cm^−1^, and of the C–N stretching peak (amide III band) at 1400 cm^−1^ and 1340 cm^−1^. Below 1300 cm^−1^ a region densely populated by several different weak signals is observed; such portion of the spectrum recalls the FT-IR features of citric acid and is thus tentatively associated to the presence of carboxylate moieties (1110 cm^−1^ and 1190 cm^−1^) [65,66]. Although the C=O stretching vibration of this group is usually found at 1700 cm^−1^, in our case it is only partially visible as a shoulder due to the dominant amide I signal. 

CDs’ optical properties, studied using both steady-state and time-resolved techniques, are presented in Figure 2. The absorbance spectrum of the sample (Figure 2a) presents a main band with a maximum at 550 nm and an additional secondary shoulder at 520 nm. Excitation at λ_exc_ = 550 nm results in an intense red emission peaked at 610 nm (Figure 2b) with a QY of (30.0 ± 0.8)%. PLE spectrum measured on the low-energy shoulder of this band (λ_em_ = 625 nm, Figure 2a) confirms how most of the absorbance spectrum is indeed responsible for this red emission. An additional component of the absorbance is revealed by the PLE measured at λ_em_ = 530 nm (Figure 2a). This band presents a maximum at 450 nm and, once excited, generates a green emission peaked at 530 nm. As often reported in the literature [67,68,69,70,71], the CDs show an excitation dependent emission (Appendix A). In particular, our CDs are able to generate a dual (green and red) radiative emission. When excited at 465 nm (i.e., the peak wavelength of the blue LED chip used for excitation in color-conversion experiments, see Section 2.10, Section 3.4 for more details), both green and red bands have approximately the same intensity (Figure 2b). This feature is relevant as enables the application of CDs as color-converting material towards white LEDs. Finally, two additional bands in the absorbance spectrum at 350 nm and 650 nm are found not to contribute to any emission. The former is probably due to n–π* transitions arising in the organic surface shell of the nanoparticles, as often reported [72,73,74], which in our case led to dark recombination pathways, while the origin of the latter faint band cannot be unambiguously assigned. 

In general, the optical bands related to the green emission observed in our sample are reminiscent of 4-hydroxy-1*H*-pyrrolo [3,4-c]pyridine-1,3,6(2*H*,5*H*)-trione (HPPT), a fluorophore which has been found to form during the synthesis of carbon nanoparticles when using the same molecular precursors; this compound, adsorbed on the CDs’ surface, has been often indicated as source for their green emission [75]. The origin of the red emission band, instead, has been previously indicated due to the presence of amino groups that, working as N dopants, are able to promote the formation of localized conjugated domains [76]; this mechanism is then able to induce a strong red-shift in the emission of blue-emitting CDs. Our case, nonetheless, differs from this example as the observed emission peak is at 610 nm while the aforementioned mechanism is able to red shift the emission up to almost 670 nm. In addition, the shape of the CDs’ excitation band (Figure 2a), composed by a main peak at 550 nm and a secondary shoulder at 520 nm, suggests that our red emission has a molecular origin. In such a case, the two peaks would correspond to transitions from different vibrational states of the formed molecule; as described for HPPT in green-emitting CDs, during the synthesis of the carbon nanoparticles these fluorescent molecules would form concurrently to the carbonaceous core and bind to its structure. A possible energy band diagram explaining the observed optical properties of Figure 2a,b is depicted in Figure 2c. These hypothesized energy levels take into account the separate optical activity of the two distinct green and red emitting fluorophores originated during the synthesis.

The emission properties of CDs were then studied by measuring their time resolved decay kinetics. The CDs solution was excited at 485 nm, and decay traces were acquired at 550 nm, 590 nm and 625 nm; the experimental data and corresponding best-fitting curves obtained from a least-squares minimization procedure are shown in Figure 2c. A bi-exponential time decay was used as model function, as detailed in the Appendix A, yielding the parameters shown in Appendix A. It is worthwhile to note how the recombination dynamics result faster at the increasing of the emission wavelength. In addition, the variations observed in both lifetimes at different emission wavelengths (Appendix A) seem indicative of a strong heterogeneity; such an evidence is not uncommon in the case of CDs [1,77,78]. For blue emitting CDs, where the emitting properties were demonstrated to be linked to a molecular fluorophore, such heterogeneity was attributed to the binding of the fluorescent species to the carbonaceous structure of CDs [79].

### 3.2. Numerical Simulations for the Design of CDs Based Nanocomposite Films

The PL spectrum of CDs (λ_exc_ = 465 nm), consisting in green and red bands of comparable intensity, strongly suggests a profitable use of such nanoparticles to operate a blue to green/red color-conversion to white light. Based on the results reported in Figure 2b, we can reasonably expect that, when CDs are embedded in an optically transparent polymeric matrix, and are excited with a suitable external source (a blue LED chip with emission peaking at 465 nm), a final spectrum covering blue, green and red regions of the visible range is generated, if the nanocomposite is loaded with an adequate content of CDs [42,80,81].

However, it is not straightforward to establish exactly how the spectroscopic properties of the CDs, as well as other characteristics of the film, such as its thickness or the concentration of CDs in the nanocomposite, affect the perceived color of light generated after color-conversion. In this sense, the process of improvement and optimization of their converting properties can be facilitated by using numerical predictions.

In general, the resulting final spectrum can be described as the sum of a green/red component (from PL of CDs) and of a blue component (from the external LED chip). More specifically, the final relative spectral density of irradiance (ϕ(λ)) can be expressed as a linear combination of that of the green/red component from PL ($\phi^{GR}\left(\lambda \right)$) and that of the blue component from the external LED source ($\phi^{B}\left(\lambda \right)$) (all algebraic steps involved in mathematical definitions of these quantities are described in detail in the Appendix A):(2)$$\phi \left(\lambda \right)=f\cdot \phi^{GR}\left(\lambda \right)+\left(1-f\right)\cdot \phi^{B}\left(\lambda \right)$$

The coefficient f in Equation (2) expresses the fraction of green/red component in ϕ(λ), i.e., the fraction of color-converted light, whereas 1−f expresses the fraction of residual blue component in the final spectrum. Specifically, in mathematical terms:(3)$$f=\frac{A^{GR}}{A^{GR}+A^{B}}$$
(4)$$1-f=\frac{A^{B}}{A^{GR}+A^{B}}$$
where $A^{GR}$ and $A^{B}$ are the integrated areas associated to PL and to external LED chip in the final spectrum (Appendix A, Equation (2): choice of normalization conditions and definitions of relative quantities).

It is worth noting that, in Equation (2), the relative spectral densities of irradiance $\phi^{GR}\left(\lambda \right)$ and $\phi^{B}\left(\lambda \right)$ only depend on the characteristic spectroscopic properties of CDs and on the selected LED source. On the contrary, the color-converting factor f is generally strongly influenced by several factors, such as: (i) the intrinsic CDs properties (i.e., their QY, that estimates their converting efficiency); (ii) the CDs concentration and thickness of the nanocomposite; and (iii) the geometrical characteristics of the device where the color-converting layer is integrated [11,82,83,84]. In fact, divergence of incident and emitted light beams and viewing angles can affect the relative intensities of blue and green/red light and hence the colorimetric properties of the final generated light.

Only in a very simplified case, i.e., considering a hypothetical geometry in which incident light is a perfectly collimated beam, generated light is collected across a 360° angle and reflection and scattering at the nanocomposite surface can be neglected; f will depend only on the nanocomposite material properties. In this simplified case, it is possible to show (detailed demonstration in provided in the Appendix A) that the resulting f (called $f_{m}$) assumes the following form:(5)$$f_{m}=\frac{c}{c+1}$$
where c expresses the dependence on the nanocomposite properties:(6)$$c=QY\left(e^{\alpha_{B}d}-1\right)$$
that are: (i) the QY of CDs embedded in the nanocomposites under irradiation from the external LED source; (ii) the nanocomposite linear absorption coefficient at the wavelength of incident blue radiation; and (iii) the film thickness (d).

$\alpha_{B}$ is proportional to CDs’ concentration in the nanocomposite. Moreover, the product $\alpha_{B}d$ is equal to the absorbance of the film at the wavelength of incident blue LED light [85]. Therefore, it can be concluded that, for a color-converting film in a device with a fixed geometry, f can be varied experimentally fabricating nanocomposites with a variable absorbance, either by regulating the thickness or the concentration of absorbing nanoparticles (i.e., the CDs loading in the nanocomposite). This remarks how physical parameters that can be easily regulated during film fabrication (CDs loading and film thickness) can be used to modulate the final spectral response of the nanocomposite (ϕ(λ)), up to eventually reaching a final spectrum that corresponds to a perceptually white color.

However, in a real device, the specific geometry of sample irradiation and light collection will affect the observed spectrum, composed of blue light diffused and transmitted from the nanocomposite and green/red light that is emitted with an intensity that has an angular dependence. Therefore, in general, due to geometrical restrictions that make it impossible to collect light across a 360° angle, the observed f is also dependent from a geometrical factor (g). In particular, it can be considered to have a form of the type:(7)$$f_{g}=g\cdot f_{m}$$

Determination of g requires the exact knowledge of several factors, such as the angles of sample irradiation and resulting light collection, the irradiated area of the sample, the angles of divergence of the blue diffused and PL emitted light, the acceptance angle and the position of the final observer [11,82,86,87,88,89]. Similar considerations also generally hold in the case of experimental measurements of colorimetric properties of surfaces, where the generated spectrum from a sample surface (also called color stimulus) is usually dependent on the measuring geometry [11]. Therefore, also determination of $f_{g}$ is very complicated, since it requires knowledge of g.

The objective appearance of the color of the nanocomposite surface can be evaluated following prescriptions of colorimetry, in terms of colorimetric parameters, such as CIE chromaticity coordinates, Correlated Color Temperature (CCT) and Color Rendering Index (CRI). Such colorimetric properties can be calculated knowing the relative spectral power distributions of the nanocomposite emitting surface or color stimulus (ϕ(λ)) [86].

In general, variations in ϕ(λ) can be brought on by modifications of $\phi^{GR}\left(\lambda \right)$, $\phi^{B}\left(\lambda \right)$ and f. However, since we are not changing the employed fluorophores nor the external blue LED source, we are particularly interested in evaluating modifications of colorimetric properties for fixed relative spectral densities of irradiance $\phi^{GR}\left(\lambda \right)$ and $\phi^{B}\left(\lambda \right)$, while f ranges from 0 to 1. Notably, for a fixed geometry, this is equivalent to assessing how experimentally controllable parameters such as $\alpha_{B}$ and d affect the final colorimetric properties of the nanocomposite. Therefore, taking f as variable, we can simulate several ϕ(λ) and then use these results to obtain simulated CIE chromaticity coordinates.

Among all the as-simulated coordinates, we are interested in finding those corresponding to a white hue. Selecting an adequate white standard illuminant and considering its chromaticity coordinates as goal point, we can quantitatively assess how distant the simulated (x, y) are from such ideal chromaticity coordinates. In particular, CIE D65 illuminant was selected, following CIE recommendations on uniformity of practice in the evaluation of whiteness of surface colors [11,86]. We simulated 1000 chromaticity coordinates, letting f vary from 0 to 1 and then the best achievable white coordinates were calculated as the simulated (x, y) values that minimize the distance with the coordinates of the reference illuminant. Thereafter, based on the calculated ${\left(x,y\right)}_{baw}$, we also identified the other correspondent minimum-distance color parameters: $f_{baw}$, $CCT_{baw}$ and $CRI_{baw}$ (Section 2.9).

In Figure 3, the graphical evaluation of the dependence of CIE coordinates (x, y) on f and the calculation of the most favorable colorimetric parameters are shown. For this calculation, $\phi^{B}\left(\lambda \right)$ is measured recording a blank LED chip spectrum in the experimental configuration described in Section 2.10, using a bare polymer reference sample without CDs on a quartz substrate (procedures of nanocomposite preparation is reported in Section 2.6), while the emission of CDs embedded in an optically transparent matrix is simulated using the PL spectrum of CDs for excitation at 465 nm (Figure 2b) to obtain $\phi^{GR}\left(\lambda \right)$.

Since the resulting ${\left(x,y\right)}_{baw}$ are very close to those of the standard D65 illuminant (Appendix A), and the resulting $CRI_{baw}$ is >80, that is an acceptable value for most white LED light bulbs [11], the outcome of the calculation indicates that CDs can effectively serve as blue to white color converting materials.

### 3.3. Preparation and Characterization of Nanocomposite Films

Color-converting nanocomposites can be prepared by uniformly dispersing the fluorescent carbon nanoparticles in a transparent polymeric matrix of PVA using water as common solvent. CD powders were mixed with the polymeric host (PVA) in aqueous medium to produce a fluorescent ink, taking advantage of the high solubility of PVA, PVP (employed for CD powder fluorophores preparation) and carbon nanoparticles in water (Section 2.5 and Section 2.6).

Then the final nanocomposite was obtained as solid-state film by depositing the ink onto a quartz substrate. To achieve a homogeneous coating with a complete surface coverage of the substrate, in-plane uniformity and low surface roughness, it is crucial to control the deposition process. Moreover, nanocomposite films with variable thickness and nanoparticle loading can be prepared by properly choosing the deposition technique, CDs concentration and deposition protocol parameters.

As discussed in Section 3.2, concentration of CDs and film thickness are important parameters in determining the fraction of incident blue light that is converted into PL emitted green/red light in Equations (3)−(6). Indeed, either an increase of nanoparticle content (leading to an increase of $\alpha_{B}$) or film thickness can turn in an increase of the fraction of converted light (f factor), thereby producing a modulation in the color of the final light. Therefore, both for drop-cast and spin-coated film, the color-converting properties of the nanocomposites were carefully controlled by properly adjusting the preparative parameters, achieving a final spectrum with chromaticity coordinates close to those of the standard white illuminant.

#### 3.3.1. Drop-Cast Films

In the ink preparation, the adjustment of CD powders and PVA concentration allows to control both final concentration of carbon nanoparticles and viscosity of the ink solution. To obtain a final film with good surface coverage and macroscopic in-plane thickness uniformity, control of viscosity and deposited ink solution volume resulted fundamental. Moreover, the evaporation rate of the solvent had to be carefully controlled (Section 2.7). In Table 1 a list of the prepared samples along with the set of parameters used for their preparation are reported.

Examination of the fabricated nanocomposites by in-plane and cross-section SEM allows to investigate more in detail the microscopic morphological features and to determine the thickness of the films. In-plane micrographs of the films (Figure 4a) show a very homogeneous deposition. Sporadically, only very small (<1 μm) agglomerates are observed onto the surface, likely due to dust or residues from the preparation and evaporation processes. The uniformity of the contrast and the absence of large surface defects suggest that carbon nanoparticles are homogeneously dispersed within the film, without visible aggregation. Cross-section SEM micrographs (Figure 4b) reveal that the thickness of the films is in the range of hundreds of μm (detailed procedure of estimation of thickness from cross-sectional SEM micrographs, called d_SEM_, is reported in Section 2.7). Interestingly, a marked variation of the final film thickness can be easily achieved by controlling the volume of the drop-cast solution, as can be observed from the comparison of the thicknesses of DC-E (Figure 4a) and DC-B (Appendix A), reported in Appendix A. In fact, the volume of the drop controls directly the amount of deposited material and hence has a direct effect on the thickness of the final film as obtained after solvent evaporation.

Optical properties of the drop-cast films have been characterized by recording absorption and steady-state PL spectra and absolute QYs (Figure 5); the latter have been measured by exciting samples at 465 nm, the same peak wavelength of the blue LED chip emission.

Since PVA is optically transparent in the visible range [37], the embedded CDs are responsible for the optical properties of the nanocomposites in that spectral region. Indeed, absorption and PL spectra (Figure 5a,b respectively) show features similar to those of CDs in aqueous solutions for excitation at 465 nm (Figure 2b). However, both absorption and PL bands are broader in the nanocomposites and the absorption maximum results redshifted at 570 nm. Such differences can be related to effects of the different local chemical environment that the CDs experience in the polymer matrix, with respect to CDs in aqueous solution. More specifically, since CDs are hydrophilic, it can be expected that, when wrapped by the polymer chains, their polar surface groups may interact with the polar pendant groups (hydroxyl and pyrrolidone moieties) of the polymers constituting the nanocomposite matrix (PVA and PVP). Additionally, in the nanocomposites’ absorption spectra, a background due to scattering of incident radiation is present, that can be attributed either to the formation of aggregates of the polymer passivated CDs (as described in Section 3.1) or to surface roughness features of the thick drop-cast films. Compared to CDs in aqueous solution, the absorption band at longer wavelengths (~670 nm) is significantly more prominent for all nanocomposite samples. Moreover, the intensity of this band relative to the maximum at 570 nm changes from sample to sample. More specifically, their ratio increases going from sample DC-A to DC-E (i.e., for decreasing film absorbance, Appendix A). Additionally, the comparison between the relative intensities of the green and red fluorescence bands in the PL spectra (Figure 5b) reveals that the intensity of the red band progressively decreases from DC-A to DC-E. Since the absorption band peaked at ~570 nm overlaps the spectral region of the red PL emission band, we can ascribe the progressive relative decrease in intensity of such red band to an inner filter effect due to photons’ re-absorption therefore occurring in the same region.

Finally, absolute QY of DC-A, DC-B and DC-C (inset of Figure 5b) resulted of ~10% (all measured QYs are within the range of 1 standard deviation), despite the difference in CDs content in the nanocomposite and thickness of the samples. For samples DC-D and DC-E a reliable QY value could not be measured because of their very low absorbance, comparable to the scattering losses at 465 nm. Nonetheless, having a weak red emission band, such samples are expected to yield poor color-converting performances. Overall, DC-A, DC-B and DC-C, with green and red emission of almost comparable intensity, represent the most suitable candidates for achieving a white spectrum through color conversion.

#### 3.3.2. Spin-Coating Deposited Films

We investigated spin-coating deposition as an alternative approach to prepare thin films with lower absorption of incident radiation. Indeed, it is established that the spin-coating technique allows to prepare films with thickness in the micron and submicron range, depending on the viscosity of the nanocomposite, with improved surface flatness compared to drop-casting [84], being such a feature essential for preventing the scattering of incident light. Also in this case, PVA and CDs concentration in the ink were varied; moreover, to modulate the film thickness the spinning speed of the first deposition step was used as an additional parameter, while both spinning speed of the second (high speed) step and deposition times were left constant for all of the prepared samples.

Two different concentrations of PVA were considered for the spin-coating deposition, namely 0.2 and 0.4 g/mL. However, the samples with the lower concentration of PVA resulted in inks with low viscosity, reflecting in films featuring Marangoni defects (such as coffee stain rings) or incomplete coverage of the quartz substrate (data not shown). Conversely, inks containing 0.4 g/mL of PVA yielded uniform films, with a homogeneous and complete coverage of the substrate surface and the absence of macroscopically observable defects (Table 2). Thus, further characterization of these films was performed by SEM, UV-Vis absorption, steady-state PL and measurement of their absolute QYs (Figure 6).

Cross-section SEM micrographs of spin-coated nanocomposites show films with average thickness of just a few μm (Table 2, Appendix A). Moreover, it is worth noting that the measured average thickness increased of ~1 μm by doubling the concentration of CD powders in the ink solution. This can be safely attributed to an increase in the ink viscosity. In fact, in most cases, there is a positive correlation between the thickness of the final film (d) and the viscosity of the spinning solution (μ), which can be explained, for example, through the Meyerhofer Model, for which $d\;\propto \mu^{1/3}$ [90]. On the other hand, tuning spinning speeds seems less effective for modulating the film thickness. Indeed, achieving an accurate control on a ~μm thick film, while preserving macroscopic uniformity, complete surface coverage and microscopic surface flatness is not trivial by spin-coating [85].

UV-Vis absorption spectra of low absorbing samples SC-A, SC-B1 and SC-B2 (Figure 6b) show interference fringes, which are strongly indicative of the high surface flatness of these samples. The fringes located in the transparency region of the spectra can be used to obtain an estimate of the film thickness (d_UV-Vis_), that results in agreement with the thickness measured from cross-section SEM micrographs (details on estimation of film thickness from UV-Vis interference fringes are provided in the Appendix A).

Interestingly, PL spectra of SC-A, SC-B1 and SC-B2 films in Figure 6c show little differences in the relative intensities of the green and red bands. SC-A displays a relatively less intense green band, while in SC-B1 and SC-B2, the intensities of the two bands are almost comparable. The weaker green band in SC-A can be explained based on photon re-absorption (Appendix A), occurring due to spectral overlap with the absorption band at 570 nm, that is almost 4 times more intense in SC-A with respect to SC-B1 and SC-B2 samples.

The QYs of spin-coating deposited films are in the range between 20 and 25% (inset of Figure 6c), about twice the value of the drop-cast ones (inset of Figure 5b). Since spin-coating deposited samples are significantly thinner and less absorbing, their QY increase may be ascribed to a limited re-absorption phenomenon across the whole emission spectrum, with respect to thicker and/or possibly more concentrated nanocomposites.

### 3.4. Evaluation of Nanocomposite Film Color Converting Performances

Based on the spectral profiles of PL emission of both the blue LED chip and the CDs for excitation at 465 nm (Figure 2b), it is possible to estimate the best achievable white coordinates obtainable using CDs fluorophores as color converters (Section 3.2). When CDs are embedded in a nanocomposite, for a fixed geometry, achievement of chromaticity coordinates close to the calculated ones can be addressed by an optimization of absorption coefficient at incident light wavelength, thickness and QY (Section 3.2). In general, for a given fabricated nanocomposite, the measured colorimetric properties (CIE chromaticity coordinates, CCT and CRI) will be more or less close to the calculated best achievable ones, depending on their actual optical properties and thickness.

Herein, colorimetric properties of all the prepared nanocomposites are measured experimentally recording the relative spectral density of irradiances with the measurement configuration described in Section 2.10 and subsequently obtained by processing the recorded spectra as reported in Section 2.8; the resulting parameters are reported in Table 3. The measured chromaticity coordinates are represented in the CIE 1931 diagram in Figure 7a.

The experimental chromaticity coordinates of samples having a PL emission spectrum with a green and red band of comparable intensity (DC-A, DC-B, DC-C, SC-A, SC-B1 and SC-B2) show a trend that may be traced back to a modulation of f (fraction of converted light) for increasing absorbance at 465 nm. This trend is similar to that simulated in Section 3.2. Deviation of the experimental coordinates from the theoretical trend can be attributed to the differences in the PL emission spectrum of CDs nanocomposites with respect to the PL spectrum of CDs in solution (Section 3.1, Figure 2b). In fact, the CDs emission spectrum was assumed as invariant for the definition of $\phi^{GR}\left(\lambda \right)$ in the simulation. However, it is worth noting that the general trend comprises both drop-cast and spin-coating deposited nanocomposites, in spite of the different deposition techniques and the more than one order of magnitude thickness difference and extinction of incident radiation (Section 3.3). This remarks how the fraction of color-converted light, determining the chromaticity coordinates and other colorimetric properties is dependent on the final properties of nanocomposite film, i.e., absorption coefficient, QY, spectral features of embedded fluorophores, thickness, being, instead, not specifically dependent on the preparation technique employed to obtain the color-converting nanocomposite.

Unlike nanocomposites having green and red emission bands of almost equal intensity, samples DC-D and DC-E show chromaticity coordinates in the light blue region of the CIE 1931 diagram. This can be explained considering that these samples have a PL emission spectrum with a less intense red component, as well as a low absorbance at 465 nm (Section 3.3). Moreover, their measured chromaticity coordinates also result out of the trend defined by the coordinates of the other samples, which can also be ascribed to the PL emission spectrum of DC-D and DC-E being significantly different from that of other CDs nanocomposites.

Finally, the samples, among those prepared, that present the colorimetric characteristics closest to those of the CIE illuminant D65, selected as reference for achieving white light, can be identified by examining the deviations of the experimentally measured properties (chromaticity coordinates, CCT and CRI) from the colorimetric properties of the standard illuminant (Figure 7b). As a result, SC-A, with chromaticity coordinates (0.30, 0.34), is the sample with the minimum coordinate distance with respect to D65. However, ΔCCT and ΔCRI for this sample are not the lowest measured, as this sample has CCT of 7100 K and CRI of 77. On the contrary, both ΔCCT and ΔCRI are the lowest for DC-C. Such sample has coordinates (0.30, 0.36), CCT of 6902 K and CRI of 82. It is worth noting that both SC-A and DC-C, despite their differences in deposition method, thickness and CDs concentration, ultimately yield a fairly similar fraction of color-converted light and somewhat akin colorimetric properties. This further demonstrates that properly adjusting the preparative conditions, both drop-casting and spin-coating deposition methods can be used to obtain color-converting nanocomposites with a satisfactory white light, although with slightly different colorimetric properties. Moreover, considering that most common commercial daylight LED-based light bulbs have a CCT of 6500 K or higher and CRI ~80 [11], we can conclude that the colorimetric features of both these samples are feasible for implementation in these types of devices.

## 4. Conclusions

To summarize, we synthesized by solvothermal approach polymer passivated CDs having a double band PL emission in the green and in the red spectral regions upon excitation with blue light. A numerical calculation method, based on the simulation of emission spectra resulting from irradiation of CDs with a commercial blue LED chip, was developed and employed to evaluate the suitability of such CDs for application in phosphor converted daylight WLEDs. Next, we incorporated the as-synthesized CDs in a PVA matrix, using spin-coating and drop-casting deposition techniques to realize solid state nanocomposite coatings, varying nanoparticle concentration over two orders of magnitude and the thickness from microns to hundreds of microns. Then the color-converting properties of such nanocomposites were experimentally measured and compared with the ideally best achievable colorimetric properties calculated with the numerical method mentioned above.

The numerically calculated best white daylight achievable from CDs resulted in a CCT of 6540 K, very close to that of the standard illuminant D65 (CCT ~6500 K) and a CRI of 82, that can be considered appropriate for the fabrication of WLEDs with good rendering of objects’ natural color [11]. On the other hand, the experimentally measured colorimetric characteristics of fabricated CDs nanocomposites showed some deviations from the simulated ideal values (and from D65 properties), which were ascribed to small variations in the spectral emission of CDs when embedded in the polymer nanocomposites, also dependent on the variable nanoparticle concentration and film morphology. The least deviations from D65 properties were found for samples SC-A and DC-C, respectively prepared by spin-coating and drop-casting. In particular, SC-A chromaticity coordinates are (0.30, 0.34), CCT = 7100 K and CRI = 77, and DC-C (0.30, 0.36), CCT = 6902 K and CRI = 82. Since the properties of both samples result suitable for daylight WLED fabrication, both employed deposition procedures (under opportune preparative parameters) appear reliable for the preparation of CDs coatings with appropriate colorimetric properties. Finally, the proposed numerical prediction of final white color properties of CDs, validated through experimental realization of samples and colorimetric property measurement, may be generalized in the future to evaluate the performances of bottom-up synthesized CDs for implementation in phosphor converted WLEDs for a given lighting application, with more time-saving procedures.

## Data Availability

The data presented in this study are available on request from the corresponding author.

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
