# Peer review of "One-Pot Synthesis of Dual Color-Emitting CDs: Numerical and Experimental Optimization towards White LEDs"

_nanomaterials, 2023, doi:10.3390/nano13030374_

Round 1
Reviewer 1 Report
In this manuscript, polymer passivated CDs with dual emission band under blue light excitation were synthesized. Further, a purposely designed numerical approach was conducted to evaluate the spectroscopic properties of CDs for application in WLEDs. Subsequently, nanocomposite coatings based on the synthesized CDs were fabricated via solution-based strategies and their color-converting properties were compared with the simulated results, to finally accomplish white light emission. This work combined numerical and experimental approach to reduce the number of experimental trials for the application of CDs in lighting area, it is novel and meaningful. Hence, this manuscript is recommended for publication in Nanomaterials. But there are still some suggestions, which can help the authors improve this manuscript.
1. The overreacted particles may not be removed after numerous cycles of washing/precipitation steps under centrifugation, leading to the CDs with large size showed in Figure S2, which may be detrimental to the fluorescence performance of CDs. Hence, filter processing is an available option for further purification of CDs.
2. The provided TEM images as Figure 1a and Figure S2 is not clear enough, and new TEM images with better quality should be provided.
3. In the FTIR spectrum of CDs, it is described that “Below 1300 cm-1 a region densely populated by several different weak signals is observed; such portion of the spectrum recall the FTIR features of citric acid and is thus tentatively associated to the presence of carboxylate moieties (1110 and 1190 cm-1)”, the FTIR spectrum of citric acid should be provided.
4. In order to fully study the optical properties of CDs, the PL spectra of CDs under different excitation wavelengths (There is a certain interval between adjacent excitation wavelengths) should be provided.
5. As shown in Figure 5, according to the nanocomposite films with different CD concentration, the CDs may possess the concentration-dependent PL emission properties, which causes the different PL spectra of nanocomposite films. Hence, it would be better to provide the PL emission spectra of CDs under different concentrations.
Reviewer 2 Report
Recommendation: Major revisions
The manuscript presents a one-pot solvothermal synthesis of polymer passivated CDs that show dual emission bands in the green and red regions under blue light excitation. The solution-based strategy was used to fabricate nanocomposite coatings based on dual color emitting CDs, and their color conversion properties were compared with simulated properties, culminating in white light emission. The manuscript can be considered for publication after some major revisions. My major concerns are listed as follow:
1. The size of the images is not uniform and adjustments need to be made
2. How photostable are the carbon dots?
3. Doesn't the aggregation of carbon dots have an effect on its light emission?
4. In Introduction section, some carbon dots related works should be cited in this paper (such as Analyst, 2021, 146 (23), 7250-7256.; Applied Physics A, 2022, 128, 356.).
Reviewer 3 Report
In this work, the authors present a type of carbon dots (CDs) that show a dual emission band (in the green and in the red regions) upon blue light excitation. Simulations are conducted to find the color converting materials that can effectively convert the blue LEDs into white emission. Detailed experiments are used to prepare the CDs and characterize its properties. In the view of my point, this work can be considered for publication. However, the following issues needs to be addressed.
1) In introduction, the authors write: “…In particular, many recent works have focused on the use of CDs in phosphor-converted white emitting LEDs (WLEDs) [5,6]. In this type of devices, UV or blue light emitted from a primary solid-state semiconductor LED chip is partially absorbed and re-emitted as photoluminescence (PL) by a phosphor material, so that a white light spectrum is produced as result [6]…” The general reference list seems a bit thin, considering the evolution in the field within the recent years. To give the readers a much broader view, recent developments related to GaN-based solid-state semiconductor LED chip, such as Nano Energy 69, 104427 (2020); Optics Express 27(12), A669 (2019); Applied Physics Letters 118, 182102 (2021); Optics Letters 47(5), 1291-1294 (2022), etc. should be added, so that the readers can be clear about the state-of-the-art of this topic.
2) To be better understood by the readers, I would suggest adding some schematic diagrams (e.g. the structure of white LEDs and the fabricating process of CDs) in the main text.
3) The mechanism that the CDs are able to generate a dual (green and red) radiative emission should be also illustrated from the aspect of energy band structure.
4) The values of QY at λexc = 465 nm in Fig. 5 are much lower than that of QY at λexc = 550 nm. What is the reason for that?
5) Why the DC-E with a higher absorbance shows a lower PL intensity as compared to the DC-C?
6) In Fig. 7, we can note that the measured and calculated colorimetric properties are not the same. Did the authors have considered to improve the simulation methods for better accuracy of prediction?
Reviewer 4 Report
In this manuscript, dual emission CDs (in the green and in the red regions) upon blue light excitation were synthesized by one-pot solvothermal synthesis. The method that combined numerical and experimental approach was applied to reduce the number of experimental trials and error steps required for optimization of CD optical properties for lighting application. The CDs doped into polymers for WLEDs. The manuscript can be published with minor revision.
1. If citric acid monohydrate was used (in manuscript, Citric acid (CA, ≥99.5%)), it is better to describe more accurately, as the molar ratio of 1.8 g of citric acid to 3.6 g of urea is 1:7.
2. After freeze-dried at low temperature, the CDs were obtained. How about the product yield for these CDs?
3. The “a.u.” in Fig. 5 is not needed in the unit for the absorbance and normalized intensity. Also, the same case for Fig. 6.
4. In the Keywords, “carbon dot” is better to use “carbon dots”, and “white emitting LEDs” to use “white LEDs”, as here “E” means emitting.
5. “In the past few years, synthesis of CDs through solvothermal treatment of various 50 types of molecular carbonaceous precursors has emerged as a very effective approach to 51 obtain multicolored emissive CDs [9,15–18,18–27].” Here, why is ref 18 repeated? Also in other sentence.
6. “More recently, similar algorithms with have been applied to color-converting components” should be “More recently, similar algorithms have been applied to color-converting components”.
7. “Simulated relative spectral density of irradiances were obtained” should be “Simulated relative spectral densities of irradiances were obtained”.
8. “In some TEM grid region” is better to use “In some TEM grid regions”.
9. “DLS analysis (Figure 1b) indicate a CDs size of 66.6 nm” should be “DLS analysis (Figure 1b) indicates a CDs size of 66.6 nm”.
10. For “The absorbance spectrum of the sample (Figure 2a) present a main band”, here present or presents?
11. “this mechanism is then be able to induce” or “this mechanism is then able to induce”?
Round 2
Reviewer 2 Report
All corrections have been made. This paper can be accepted.